# Amyloid Beta-Mediated Changes in Synaptic Function and Spine Number of Neocortical Neurons Depend on NMDA Receptors

**DOI:** 10.3390/ijms22126298

**Published:** 2021-06-11

**Authors:** Michaela K. Back, Sonia Ruggieri, Eric Jacobi, Jakob von Engelhardt

**Affiliations:** Institute of Pathophysiology, Focus Program Translational Neuroscience (FTN), University Medical Center of the Johannes Gutenberg, University Mainz, 55128 Mainz, Germany; michaela.back@uni-mainz.de (M.K.B.); rugsonia@uni-mainz.de (S.R.); eric.jacobi@uni-mainz.de (E.J.)

**Keywords:** Alzheimer’s disease, 5xFAD, Amyloid beta, NMDAR, somatosensory cortex

## Abstract

Onset and progression of Alzheimer’s disease (AD) pathophysiology differs between brain regions. The neocortex, for example, is a brain region that is affected very early during AD. NMDA receptors (NMDARs) are involved in mediating amyloid beta (Aβ) toxicity. NMDAR expression, on the other hand, can be affected by Aβ. We tested whether the high vulnerability of neocortical neurons for Aβ-toxicity may result from specific NMDAR expression profiles or from a particular regulation of NMDAR expression by Aβ. Electrophysiological analyses suggested that pyramidal cells of 6-months-old wildtype mice express mostly GluN1/GluN2A NMDARs. While synaptic NMDAR-mediated currents are unaltered in 5xFAD mice, extrasynaptic NMDARs seem to contain GluN1/GluN2A and GluN1/GluN2A/GluN2B. We used conditional GluN1 and GluN2B knockout mice to investigate whether NMDARs contribute to Aβ-toxicity. Spine number was decreased in pyramidal cells of 5xFAD mice and increased in neurons with 3-week virus-mediated Aβ-overexpression. NMDARs were required for both Aβ-mediated changes in spine number and functional synapses. Thus, our study gives novel insights into the Aβ-mediated regulation of NMDAR expression and the role of NMDARs in Aβ pathophysiology in the somatosensory cortex.

## 1. Introduction

Alzheimer’s disease (AD) is a neurodegenerative disease defined by the occurrence of amyloid beta (Aβ) plaques and tau tangles in the brain. Cognitive impairment, an early symptom in AD, occurs already before Aβ plaques start to build up. In line with this, AD progression correlates better with the amount of soluble Aβ than with the number of Aβ plaques [1]. Interestingly, load of soluble Aβ, Aβ plaques, and tangle formation vary strongly between brain areas in mouse models and human patients [2,3]. Although memory deficits occur as early symptoms in AD patients, Aβ plaques occur in the hippocampus, a brain area involved in memory formation, later than in the neocortex [3,4,5,6]. In line with these findings from AD patients, the 5xFAD AD mouse model shows spine loss and neuron loss in the neocortex earlier than in the hippocampus [7]. However, the reason for the region-specific differences in Aβ-toxicity is not known.

Memantine, an open-channel blocker of *N*-methyl-D-aspartate receptors (NMDARs) improves cognitive abilities in moderate-to-severe AD [8,9]. In addition, several studies had suggested that NMDARs play a role in Aβ-toxicity in rodent AD models [10,11,12,13]. However, the contribution of NMDARs to Aβ-toxicity differs strongly between studies. Thus, NMDAR activation is required for Aβ-mediated spine loss in cultured neurons [12,14,15,16] and brain slices [17,18,19,20,21] of AD model mice. In contrast, we and others found that NMDARs are dispensable for the Aβ-mediated reduction in spine number of hippocampal neurons in adult mice. Developmental changes and region-specific differences in expression of NMDAR subtypes may account for the diverse roles of NMDARs in Aβ-toxicity and may be the reason for brain region-specific differences in susceptibility to Aβ-toxicity. Thus, different NMDAR subunits regulate spine formation and stability in the developing and adult mouse brain [22]. Especially, the GluN2 subunits are key factors during development, such that incorporation of GluN2A stabilizes synapses and GluN2B destabilizes synapses, allowing the formation or retraction of spines [22].

NMDARs are composed of four subunits, of which two are GluN1 subunits and two are combinations of GluN2(A-D) or GluN3(A or B) subunits [23,24]. Neurons of the developing forebrain express predominantly GluN1 and GluN2B subunits and much less GluN2A, GluN2C, and GluN2D [25,26,27,28,29]. The GluN2A subunits are upregulated with development such that diheteromeric GluN1/GluN2A, GluN1/GluN2B, and triheteromeric GluN1/GluN2A/GluN2B-containing NMDARs are the most abundant forms in the adult forebrain [30,31]. The expression of NMDARs with different subunit compositions results in diverse NMDAR functions [32]. For example, GluN1/GluN2B-containing NMDARs display considerably slower decay kinetics than GluN1/GluN2A-containing NMDARs [33]. This explains why overactivation of GluN1/GluN2B-containing NMDARs induces a stronger Ca^2+^-mediated excitotoxicity than the overactivation of GluN1/GluN2A-containing NMDARs [34]. Differential expression [35] or regulation [36,37] of the subunits GluN2A and GluN2B might therefore account for brain region-specific susceptibility to Aβ-toxicity.

The mechanisms of how NMDARs contribute to Aβ-toxicity are complex and still not completely understood. Aβ may influence NMDAR activation/function by directly interacting with the channels [38], by augmenting ambient glutamate levels [39,40], or by influencing NMDAR subunit expression and subcellular localization [11,12,41]. Redistribution of NMDAR expression on the cell surface may contribute to toxicity as shown for other neurodegenerative diseases [42], because the activation of synaptic NMDARs stimulates pro-survival signaling, whereas the activation of extrasynaptic NMDARs induces neuron apoptosis [43,44] (but see also: [45]).

To test whether specific expression and/or regulation of NMDAR subunits contribute to the early neuronal Aβ-toxicity in the neocortex, we investigated NMDAR-mediated currents in neocortical neurons with virus-mediated Aβ-overexpression as well as in neocortical neurons of the 5xFAD mouse model. In contrast to the previously observed downregulation of NMDARs in hippocampal neurons [12,46], we observed little change in the amplitude of synaptic and extrasynaptic NMDAR-mediated currents in neocortical neurons of 5xFAD mice. However, we found changes in the decay and deactivation kinetics consistent with an upregulation of extrasynaptic GluN2B-containing NMDARs. The contribution of NMDARs to Aβ-toxicity in adult neocortical neurons was investigated using conditional knockout of GluN1, GluN2A, or GluN2B. Three weeks of Aβ-overexpression reduced the number of functional synapses and increased the number of spines. In contrast, spine number was reduced in neocortical neurons of 6-months-old 5xFAD mice. The Aβ-mediated change in spine number depended on the presence of NMDARs. Finally, deletion of NMDARs per se reduced spine number. Thus, our study gives novel insights into the regulation and role of NMDARs in AD pathophysiology.

## 2. Results

### 2.1. 5xFAD Mice Show a High Intracellular Aβ Burden and Aβ Plaques in the Somatosensory Cortex

To investigate Aβ-toxicity in the cortex, we used the 5xFAD mouse model. This well-established mouse model shows early and aggressive Aβ accumulation [47]. Previously, we had observed that the number of synapses and spines were reduced in dentate gyrus granule cells of 5xFAD mice at an age of 12-months, but unaltered in 6-months-old 5xFAD mice [46]. Given that Aβ-expression, onset, and severity of Aβ-pathology differs between brain regions with early onset and severity in the cortex of 5xFAD mice [7,47], we investigated whether signs of Aβ-overexpression can be observed in the somatosensory cortex of 6-months-old 5xFAD mice. Aβ immunofluorescence staining using the 6E10 antibody showed strong Aβ signals in the somatosensory cortex of 5xFAD mice (Figure 1). This antibody detects amino acid residues 1–16 of the Aβ protein. Thus, it can be used to stain soluble forms of Aβ as well as Aβ plaques. The Aβ-staining was found mainly intracellularly in neurons of the somatosensory cortex of 6-months-old 5xFAD mice accompanied by a smaller amount of extracellular Aβ plaques (Figure 1). No Aβ staining was observed in aged-matched wildtype (WT) mice (Figure 1).

### 2.2. NMDAR Subunit Expression Is Regulated by Aβ

Aβ toxicity may occur in the cortex earlier than in other brain areas because of regional differences in the expression of NMDARs. In addition, Aβ may affect NMDAR expression in cortical neurons, which in turn could lead to alterations in synapse function [12,14] and ultimately to cell death [48]. Since activation of extrasynaptic NMDARs was shown to trigger cell death pathways whereas activation of synaptic NMDARs reduces cell death [43,49,50], but see [51,52,53,54], we recorded synaptic and extrasynaptic NMDAR-mediated currents in layer 5 pyramidal cells of the somatosensory cortex in 6-months-old 5xFAD mice.

The α-amino-3-hydroxy-5-methyl-4-isoxazolepropionic acid receptor (AMPAR)/NMDAR ratio) was not different in somatosensory cortex pyramidal cells from 5xFAD mice compared to WT mice (Figure 2a). Given that AMPAR-mediated currents were not significantly affected in pyramidal cells of the somatosensory cortex from 5xFAD mice, evidenced by unaltered miniature excitatory postsynaptic currents (mEPSCs) (see below), these data suggest that Aβ does not influence the expression of synaptic NMDARs in pyramidal cells. NMDAR current decay was also similar in 5xFAD and WT neurons (Figure 2b). Considering that the subunits GluN2A-D strongly influence deactivation kinetics [35], the unaltered decay suggests that Aβ does not affect NMDAR composition in pyramidal cells of the somatosensory cortex. Of note, the decay time constant (~29 ms) indicates that synaptic NMDARs are pure GluN1/GluN2A receptors [31,55].

Extrasynaptic NMDARs were investigated by ultra-fast application of glutamate onto nucleated patches of pyramidal cells of the somatosensory cortex of 6-months-old 5xFAD mice. This analysis revealed no difference in the peak amplitude of NMDAR-mediated currents (Figure 2c), but an increase in the deactivation time constant (*t*-test; *p* = 0.0084) (Figure 2d). These data suggest that the number of extrasynaptic NMDARs is unaltered in 5xFAD mice, but that there is an increase in the contribution of slow deactivating GluN2B-containing NMDARs [31]. However, the deactivation time constant of extrasynaptic NMDAR-mediated currents (~35 ms) in pyramidal cells of WT mice suggests that also extrasynaptic NMDARs are by and large GluN1/GluN2A receptors with a small contribution of triheteromeric GluN1/GluN2A/GluN2B receptors [31,55]. In addition, the deactivation of extrasynaptic NMDARs in 5xFAD mice is still very fast (52 ms), suggesting that the majority of NMDARs comprise GluN1/GluN2A receptors and GluN1/GluN2A/GluN2B with very little, if at all, contribution of diheteromeric GluN1/GluN2B receptors.

### 2.3. NMDAR Subunits Are Involved in Mediating Spine Loss in Somatosensory Cortex Pyramidal Cells of 5xFAD Mice

Previous studies suggested that NMDARs and in particular those containing the GluN2B subunit mediate Aβ-toxicity in hippocampal neurons. We asked whether NMDARs play a similar role in cortical neurons and investigated the involvement of the GluN1, GluN2B, and GluN2A subunits in Aβ-mediated changes on dendritic spine number and synapse function.

Spine number was decreased by about 12% in pyramidal cells of the somatosensory cortex of 6-months-old 5xFAD mice when compared to control mice (*t*-test; *p* = 0.0443) (Figure 3b,c). To investigate the requirement of NMDARs for this effect, we deleted GluN1, GluN2B, or GluN2A by stereotactc injection of Cre-recombinase expressing rAAVs into the somatosensory cortex of conditional NMDAR knockout mice [46]. Spine number was not different between neurons with GluN1 deletion (GluN1^−/−^) and neurons of 5xFAD mice with GluN1 deletion (5xFAD/GluN1^−/−^) (Figure 3b), suggesting that NMDARs are required for the Aβ-mediated spine loss. Similarly, there was no difference in spine number of somatosensory cortex pyramidal cells with GluN2B deletion (GluN2B^−/−^) and cells of 5xFAD mice with GluN2B deletion (5xFAD/GluN2B^−/−^) as well as in cells with GluN2A deletion (GluN2A^−/−^) and cells of 5xFAD mice with GluN2A deletion (5xFAD/GluN2A^−/−^) (Figure 3b). There were 0.18/µm fewer spines in pyramidal neurons of 5xFAD mice than in pyramidal neurons of WT mice (3c). There was a trend to a smaller decrease in spine number in neurons with GluN1, GluN2B, and GluN2A deletion (Δ spine number between GluN1^−/−^ and 5xFAD/GluN1^−/−^: 0.04/µm; Δ spine number between GluN2B^−/−^ and 5xFAD/GluN2B^−/−^: 0.02/µm; Δ spine number between GluN2A^−/−^ and 5xFAD/GluN2A^−/−^: 0.07/µm). The direct comparison of the change in spine number indicates that GluN2B- and GluN2A-containing NMDARs may play a role in Aβ-toxicity in cortical neurons. Of note, deletion of GluN1, GluN2B, and GluN2A per se reduced spine number in somatosensory cortex pyramidal cells (i.e., compared to spine number in WT mice), suggesting that NMDARs control the number of spines in cortical neurons.

### 2.4. NMDARs Are Involved in Mediating the Decrease in Functional Synapse Number in Pyramidal Cells of the Somatosensory Cortex of 5xFAD Mice

Glutamatergic synapses are located on dendritic spines. We therefore investigated whether the spine loss was accompanied by changes in functional synapse number.

Synaptic function was analyzed by recording mEPSCs from somatosensory cortex pyramidal cells of 6-months-old control and 5xFAD mice. mEPSC frequency is commonly used as an estimation of the number of functional synapses of a neuron, whereas mEPSC peak amplitude gives an estimation of the number of AMPARs per synapse [56].

mEPSC frequency was not altered in 5xFAD mice compared to WT controls (Figure 4d). There was no difference in mEPSC frequency between neurons from WT and 5xFAD mice with GluN1 (red bar vs. grey bar) or GluN2A deletion (pink bar vs. lilac bar) (Figure 4d). A subtle increase of mEPSC frequency was found in neurons of 5xFAD mice with GluN2B deletion (green bar vs. dark grey bar; Mann-Whitney test: * *p* = 0.0398; Figure 4d). This increase in mEPSC frequency was not significantly different to the change in mEPSC frequency in neurons without NMDAR deletion (Kruskal-Wallis: ** *p* = 0.0029; Figure 4e). Overall, in 6-months-old 5xFAD mice, somatosensory cortex pyramidal cells do not show deficits in functional synapse number. To see whether functional synapse number was affected in older 5xFAD, when the disease progressed further, we recorded mEPSCs in 1-year-old pyramidal cells of the somatosensory cortex. At this age, we had previously shown a loss of functional synapse number in dentate gyrus granule cells (Mueller et al., 2018). However, mEPSC frequency and peak amplitude were not affected in 1-year-old somatosensory cortex pyramidal cells from 5xFAD mice (Appendix A Appendix A). mEPSC amplitude was slightly increased in neurons of 6-months-old 5xFAD mice (Mann-Whitney test: * *p* = 0.0428; Figure 4f), but not significantly affected in somatosensory cortex pyramidal cells of 5xFAD mice with deletion of GluN1, GluN2B or GluN2A (Figure 4f). However, the changes in mEPSC amplitudes in neurons without and with NMDAR subunit deletion were not significantly different from each other (Figure 4g).

### 2.5. Neuronal Excitability of Somatosensory Cortex Pyramidal Cells Is Not Altered in 5xFAD Mice

Aβ-mediated changes of intrinsic properties of neurons [57,58,59] were discussed to contribute to neuronal hyperactivity in AD patients and mouse models [60]. We therefore investigated whether action potential properties and firing was altered in pyramidal cells in the somatosensory cortex of 6-months-old 5xFAD mice.

Intrinsic properties including action potential threshold, amplitude, duration, input resistance, afterhyperpolarization (Figure 5b), as well as firing behavior measured by action potential firing frequency; early and late adaptation of firing were not different between pyramidal cells from WT and 5xFAD mice (Figure 5d). Thus, 6-month overexpression of Aβ in 5xFAD mice did not influence active and passive electrophysiological properties of pyramidal cells in the somatosensory cortex and suggests that Aβ-toxicity is comparably mild at this age despite the presence of intracellular Aβ accumulation and extracellular Aβ plaques.

### 2.6. Spine Number Is Increased after Short-Term Overexpression of Aβ in the Somatosensory Cortex

The deletion of NMDARs per se reduced spine number in somatosensory cortex pyramidal cells in 6-months-old WT mice (Figure 3b), which may obscure effects of Aβ on spine number. It was previously shown that deletion of GluN2B reduces spine number also in CA1 neurons [61]. However, the reduction in CA1 neurons is more subtle, which may be explained by the much shorter period of NMDAR deletion (2–3 weeks) in that study compared to the 3-month deletion in our study. We therefore investigated the involvement of NMDARs on Aβ-toxicity in an additional model in which we overexpressed Aβ and deleted NMDARs in somatosensory cortex pyramidal cells for a much shorter time period.

To this end, we induced the overexpression of Aβ by injecting into the somatosensory cortex of adult WT mice an rAAV (rAAV-CaMKII-CT100(I716F)-T2A-tdTom), which expresses the penultimate Aβ precursor CT100(I716F), harboring a mutation that increases the Aβ42/Aβ40 ratio [62] and tdTomato to detect infected neurons. NMDAR subunit deletion was induced by co-injection of an rAAV coding for Cre-recombinase into conditional NMDAR knockout mice. Functional and anatomical analyses were performed 3 weeks after virus-injection [46].

Anatomical analysis of pyramidal cells of the somatosensory cortex revealed that 3 weeks Aβ-overexpression increased spine number when compared to control cells (Mann-Whitney test: *** *p* = 0.0002) (Figure 6b). Deletion of GluN1 or GluN2B for 3 weeks did not affect spine number of somatosensory cortex pyramidal cells. This is in contrast to the reduction of spine number after 3-month deletion of GluN1 or GluN2B (Figure 3b). Aβ-overexpression did not significantly increase spine number in neurons with GluN1 or GluN2B deletion. The direct comparison of the Aβ-induced change in spine number revealed that deletion of the GluN1 subunit significantly prevented the Aβ-mediated increase in spine number (Figure 6c). In contrast, there was a trend for an Aβ-mediated increase in spine number in neurons with GluN2B deletion, which was not significantly different to the Aβ-mediated increase in control neurons (Figure 6c). Thus, the Aβ-mediated formation of new synapses was regulated by NMDARs.

We previously showed that functional synapse number was reduced, while spine number was unaffected in dentate gyrus granule cells overexpressing Aβ, suggesting an increase in immature synapses that do not contain AMPARs [46].

### 2.7. NMDAR Subunits Are Involved in Aβ-Mediated Changes in Functional Synapse Number

AMPAR-mediated mEPSCs were recorded in order to study synapse functionality after 3 weeks of virus-mediated Aβ-overexpression. mEPSC frequency was reduced by about 42% (0.89 Hz) in Aβ-overexpressing pyramidal cells (Mann–Whitney test: *** *p* = 0.0003) (Figure 7d), consistent with a reduction in functional synapse number. Importantly, Aβ-overexpression did not reduce mEPSC frequency in neurons with deletion of GluN1 or GluN2B. Direct comparison of the change in mEPSC frequency revealed that the Aβ-mediated change in mEPSC frequency was significantly different in neurons without NMDAR deletion from neurons with GluN2B deletion (Figure 7e). mEPSC peak amplitudes were not different between genotypes (Figure 7f,g).

The decrease in mEPSC frequency shows that Aβ-overexpression reduces functional synapse number in pyramidal cells of the somatosensory cortex, which is prevented by deletion of the GluN1 or GluN2B. This indicates that GluN2B-containing NMDARs are involved in the Aβ-mediated reduction in functional synapse number.

## 3. Discussion

The progression of AD pathology follows distinct patterns in the brains of AD patients and also in AD mouse models [2,3]. The neocortex is a brain region that is involved very early in AD in human patients [2] and in AD model mice. Thus, cortical plasticity in the somatosensory cortex is more impaired compared to the hippocampus in 6-months-old 5xFAD mice [63]. Consistently, spine loss—a common pathology in AD mouse models and human AD patients—is present in pyramidal cells of the somatosensory cortex but not in dentate gyrus granule cells of 6-months-old 5xFAD mice (this study and [7,46]). Dentate gyrus granule cells are not resistant to Aβ-toxicity, but spine loss commences later and is significantly smaller in dentate gyrus granule cells of 1-year-old 5xFAD mice [46].

Memantine, an NMDAR open-channel blocker, is widely used to treat moderate-severe AD in human patients [8,9]. In addition, there is ample evidence from mouse studies that NMDARs and, in particular, the slow-deactivating GluN2B-containing NMDARs are involved in Aβ-toxicity [12,15,17,21,64]. The region-specific differences in susceptibility for Aβ-toxicity may therefore result from differential NMDAR expression. Knowledge of the brain-region specific composition and contribution of the NMDAR subunits to AD pathophysiology could be important for the development of novel AD treatments. For example, antagonists that are specific for AD-relevant NMDAR compositions might be more efficient than memantine without the cost of stronger side effects. Antagonists that are more specific for triheteromeric NMDARs could be more beneficial than memantine for the treatment of AD patients considering the upregulation of triheteromeric NMDARs at extrasynaptic sites in 5xFAD somatosensory pyramidal cells. The composition of NMDARs can be estimated from deactivation kinetics. Thus, deactivation is fast for NMDARs containing the GluN2A subunit, slower for NMDARs containing the GluN2B or GluN2C subunit, and very slow for NMDARs containing the GluN2D subunit [32]. Synaptic NMDAR-mediated currents of somatosensory cortex pyramidal cells decay very fast. The decay time constant of ~35 ms is similar to the deactivation time constant of diheteromeric GluN1/GluN2A NMDARs (23 ms; [65]), which indicates that the contribution of NMDARs containing the GluN2B, GluN2C or GluN2D subunit is negligible in synapses of somatosensory cortex pyramidal cells.

Importantly, the subcellular localization of NMDARs is relevant for the role in toxicity [49]. Thus, synaptic NMDARs mediate pro-survival signaling via CREB activation, whereas the activation of extrasynaptic NMDARs induces cell-death pathways [43]. Excessive activation of extrasynaptic NMDAR signaling, partially due to an increase in extrasynaptic NMDAR number, was shown in different neurodegenerative diseases, including AD [42,66]. The detrimental role of extrasynaptic NMDARs in neurodegenerative diseases has been connected with a preferential extrasynaptic expression of GluN2B-containing NMDARs, which display slow deactivation kinetics [35]. We therefore tested whether the expression of extrasynaptic NMDARs with particular slow deactivation kinetics could explain the sensitivity of somatosensory cortex pyramidal cells to Aβ-toxicity. However, the deactivation of extrasynaptic NMDARs was also very fast (35 ms) and consistent with the expression of mostly diheteromeric GluN1/GluN2A NMDARs. In fact, decay of synaptic NMDAR-mediated currents and deactivation of extrasynaptic NMDAR-mediated currents was much faster than in dentate gyrus granule cells [46]. The decay time constant (~63 ms) of synaptic NMDAR-mediated currents and deactivation time constant (~75 ms) of extrasynaptic NMDAR-mediated currents in dentate gyrus granule cells is consistent with the expression of triheteromeric GluN1/GluN2A/GluN2B NMDARs or a mixture of diheteromeric GluN1/GluN2A and GluN1/GluN2B NMDARs [46]. Moreover, the amplitude of extrasynaptic NMDAR-mediated currents was considerably lower in pyramidal cells of the somatosensory cortex (~36 pA) than in dentate gyrus granule cells (~125 pA; [46]), indicating that the number of extrasynaptic NMDARs is much smaller in pyramidal cells than in dentate gyrus granule cells. Thus, differences in the expression of NMDARs unlikely explain the higher susceptibility of cortical neurons than hippocampal neurons to Aβ-toxicity.

However, the slower deactivation of extrasynaptic NMDAR-mediated currents in pyramidal cells of 5xFAD mice than in WT mice suggests that Aβ changes the composition of NMDARs by increasing the number of receptors containing the GluN2B subunit. Of note, the deactivation time constant (52 ms) is close to that of triheteromeric GluN1/GluN2A/GluN2B NMDARs (78 ms; [31]). This indicates that extrasynaptic NMDARs in pyramidal cells of the somatosensory cortex in 5xFAD mice are a mixture of triheteromeric GluN1/GluN2A/GluN2B and diheteromeric GLuN1/GluN2A NMDARs. Considering that the activation of extrasynaptic GluN2B-containing NMDARs induces cell death [43], the replacement of fast GluN1/GluN2A NMDARs by slower deactivating GluN2B-containing triheteromeric NMDARs may contribute to Aβ-toxicity in pyramidal cells of the somatosensory cortex. The GluN1 gene can be alternatively spliced into eight different isoforms [67]. These splice variants are differently expressed in different brain areas and determine NMDAR function, gating, and kinetics [68,69,70,71,72,73,74,75]. Thus, another possible explanation for the increase in the deactivation time constant could be the incorporation of NMDARs containing GluN1 subunits that lack the alternatively spliced exon 5 [70]. To our knowledge, there is no data available on disease-induced alterations on the expression of GluN1 subunit.

In contrast to the findings in pyramidal cells of the somatosensory cortex, there is no increase in the decay or deactivation time constants in dentate gyrus granule cells of 5xFAD mice [46]. However, Aβ decreased the amplitude of synaptic and extrasynaptic NMDAR-mediated currents by nearly 50% in dentate gyrus granule cells [46]. A downregulation of NMDARs may possibly protect dentate gyrus granule cells from NMDAR-dependent Aβ pathologies. Thus, the data suggest that regional differences in susceptibility to Aβ-toxicity do not result from differential expression of NMDAR subunits but more likely from region-specific regulations in NMDAR expression.

Spine number was reduced, as expected [7], in pyramidal cells of the somatosensory cortex of 6-months-old 5xFAD mice. Interestingly, 3 weeks Aβ overexpression increased spine number in pyramidal cells of the somatosensory cortex. Although not significant, there was a trend to increase spine numbers also in dentate gyrus granule cells after 3 weeks of Aβ overexpression. An increased spine number was also observed by others after overexpression of full-length APP [76]. The most likely explanation for opposite effects on spine number is the difference in the duration of Aβ overexpression. Chronic Aβ overexpression is known to be toxic [77,78]. In contrast, the model with short-time Aβ overexpression may reveal the physiological role of Aβ or the APP intracellular domain (AICD) on synaptic activity and morphology [79,80,81,82,83]. AICD-proteolytic products were shown to translocate to the nucleus where they can activate transcription factors and thus alter the synaptic function [83,84,85,86]. Indeed, APP itself, as well as its homologues and cleavage products, affect dendritic spine numbers [76,87,88,89,90].

We deleted GluN1 and GluN2B by virus-mediated expression of Cre-recombinase in conditional NMDAR subunit knockout mice to investigate the role of NMDARs on the Aβ-induced changes in spine number and synapse function. Interestingly, NMDARs are required for the increase in spine number after 3-week Aβ overexpression as well as the decrease in spine number in 5xFAD mice. Deletion of the GluN2B subunit was not sufficient to prevent the increase in spine number after 3-week Aβ overexpression, perhaps indicating that GluN2A-containing NMDARs mediate this effect.

The interpretation of the requirement of NMDARs for the effect of Aβ on spine number is somewhat compromised by the fact that deletion of NMDARs alone strongly reduces spine number after NMDAR subunit deletion for 12 weeks in 6-month old mice, but not after 3 weeks in 3-month old mice. Thus, the duration of NMDAR absence and/or the age of the animal seem to be critical. Spines can be classified as transient (lifetime of minutes to hours) and persistent (lifetime of days to months) [91,92]. The largest fraction of spines are of the persistent type in the somatosensory cortex of adult mice [91,93]. The slow turnover explains therefore that spine loss is present 12 weeks but not 3 weeks after NMDAR deletion. It is known that NMDARs influence synapse development and stability [22]. However, the role of NDMARs for spine number appears to be region specific. Thus, deletion of NMDARs does not reduce spine number in dentate gyrus granule cells [46]. Spine number is reduced by deletion of GluN2B but not of GluN1 or GluN2A in CA1 neurons [61]. The effect appears to be smaller than in pyramidal cells of the somatosensory cortex, which may be explained by the different cell type or by the shorter period of GluN2B absence (3 weeks) in Ca1 pyramidal cells in the study of Gray and colleagues compared to the 12 weeks absence in pyramidal cells in our study. Finally, deletion of GluN2B reduces spine number also in CA3 pyramidal cells [94]. The decreased spine number in neurons with GluN2B deletion may be surprising as the findings from electrophysiological recordings suggested that pyramidal cells of the somatosensory cortex express if at all only very few GluN2B-containing NMDARs. However, one has to keep in mind that the GluN2B subunit is downregulated with development [35]. We deleted GluN2B in 3-months-old mice, 12 weeks prior to functional and morphological analysis. Thus, it is possible that the effect of GluN2B deletion on spine number results from the long-time absence of GluN2B, which includes also an age-period at which pyramidal cells of the somatosensory cortex express GluN2B.

How could the requirement of NMDARs for the effect of Aβ on spine number be explained? Considering the influence of NMDARs on spine number and stability [22,95], one possibility would be that Aβ downregulates NMDAR number in pyramidal cells of 5xFAD mice resulting in reduced NMDAR activation. However, results of electrophysiological analyses with unaltered amplitude and slowing of NMDAR-mediated currents speak against this hypothesis. Reduced glutamate re-uptake with increased ambient glutamate levels and therefore overactivation of NMDARs was discussed as a mechanism of how Aβ induces toxicity [40]. Increased ambient glutamate may augment the activation of extrasynaptically NMDARs in particular [42]. Although the question of whether synaptic and extrasynaptic NMDARs play differential roles in survival and cell-death signaling is debated [96], there is ample evidence that their activation induced different intracellular signaling cascades [44,97,98,99,100,101,102]. Thus, it is also possible that synaptic and extrasynaptic NMDARs play different roles in controlling spine number. If that is the case, an increased activation of extrasynaptic NMDARs via glutamate spillover in 5xFAD mice could lead to decreased spine numbers, which would not necessarily be contradictory to the finding of decreased spine number in neurons with genetic deletion of synaptic and extrasynaptic NMDARs. The requirement of NMDARs for Aβ-toxicity was shown in several studies. However, it is of course also possible that the influences of Aβ and NMDARs on spine number are independent of each other, but that Aβ-overexpression and NMDAR deletion induce or inhibit the same intracellular signaling cascades. In this case, an influence of Aβ on spine number might be masked in neurons with a spine number reduction due to NMDAR deletion.

Short-term Aβ overexpression decreased functional synapse number in pyramidal cells of the somatosensory cortex as evidenced by the decreased mEPSC frequency. Functional synapse number was also reduced by Aβ in dentate gyrus granule cells [46]. This influence depended in both cell types on the presence of NMDARs. A reduction in functional synapse number may seem at odds with the increase in spine number in cells with short-term Aβ overexpression. However, it can be explained by the increased formation of immature spines that do not contain AMPARs. These immature synapses, which can be converted into functional synapses by the insertion of AMPARs [103], could be built in order to compensate for the loss of functional synapses.

In conclusion, our study shows alterations in NMDAR expression and function in the somatosensory cortex of Aβ over-expressing mice. NMDARs were involved in functional and structural changes observed after 3 weeks of virus-mediated Aβ over-expression and in the 5xFAD mouse model. These data show a different impact of NMDAR expression and function in the somatosensory cortex compared to the role in the hippocampus shown in a previous study performed in the dentate gyrus [46]. Therefore, we argue that future investigation of the role of Aβ and also Tau in AD should be done with respect to region-specific differences in putative pathophysiological mechanisms. This may be relevant for the development of drugs targeting specific NMDAR combinations that are predominant in brain regions particularly vulnerable to Aβ-toxicity. Thus, early treatment with specific antagonists that act on triheteromeric (GluN1/2A/2B) NMDARs in the neocortex might be more efficient and might have fewer side-effects than broad blockade of all NMDARs by memantine.

## 4. Materials and Methods

### 4.1. Animals

All mouse experiments and breedings were carried out in compliance with the German Animal Welfare Act and the state investigation office of Rhineland Palatinate and Baden-Wuerttemberg. All procedures followed the “Principles of laboratory animal care” (NIH publication No. 86–23, revised 1985).

Mice were housed in groups of up to four animals, and food and water were offered ad libitum. Mice, in which the *grin1* and *grin2b* genes are flanked by loxP site (GluN1^fl/fl^ [104] and GluN2B^fl/fl^ [105] and 5xFADxGluN1^fl/fl^/GluN2B^fl/fl^ lines) were used to study the effect of NMDAR subunit deletion in Alzheimer’s disease (as described in [46]). Only female 5xFAD mice were used for experiments, since male and female 5xFAD show significant differences in Aβ plaque load [106]. For experiments in which Aβ overproduction was mediated via rAAVs, data obtained from female and male mice were pooled.

Littermates were used as controls in all experiments. In experiments with virus-mediated Aβ overexpression, we used GluN1^fl/fl^ and GluN2B^fl/fl^ mice injected with a tomato-expressing rAAV as control (for detail see below). We verified before that the injection of the tomato and Cre-expressing rAAVs as well as the insertion of the loxP sites per se do not alter the expression of AMPA and NMDARs (own unpublished data).

For experiments with the 5xFAD line, we injected the tomato-expressing rAAV in 5xFADxGluN1^fl/fl^ and 5xFAD/GluN2B^fl/fl^ mice (termed 5xFAD in this paper) and littermate GluN1^fl/fl^ and GluN2B^fl/fl^ mice (termed WT in this paper).

### 4.2. Stereotactic Injection of rAAVs

rAAVs were produced as previously described [46]. The following rAAVs were stereotactically injected trough a thin glass capillary into the somatosensory cortex (anteroposterior, −2.5 mm; mediolateral, + 3.5 mm; dorsoventral, −0.5 mm according to Bregma): rAAV-CaMKII-tdTom (control cells), rAAV-CaMKII-CT100(I716F)-T2A-tdTom (for CT100(I716F)-overexpression), rAAV-Syn-Cre-T2A-GFP (for NMDAR subunit deletion), and rAAV-Syn-Cre-T2A-GFP + rAAV-CaMKII-CT100(I716F)-T2A-tdTom (for NMDAR subunit deletion and CT100(I716F)-overexpression).

### 4.3. Preparation of Acute Slices

Ice-cold slicing solution (212 mM sucrose, 26 mM NaHCO_3_, 1.25 mM NaH_2_PO_4_, 3 mM KCl, 0.2 mM CaCl_2_, 7 mM MgCl_2_, and 10 mM glucose) was trans-cardially perfused in deeply anesthetized mice (3% isoflurane). Whole mouse brains were carefully extracted and cut into 250-µm-thick coronal mouse brain slices submerged in slicing solution with the help of a tissue slicer (VT1200 S, Leica, Wetzlar, Germany).

Acute slices were stored in a holding chamber filled with 37 °C artificial cerebral spine fluid (ACSF: 125 mM NaCl, 25 mM NaHCO_3_, 1.25 mM NaH_2_PO_4_, 2.5 mM KCl, 2 mM CaCl_2_, 1 mM MgCl_2_, and 25 mM glucose) for 15 min and subsequently cooled down to RT. Slices were used for electrophysiological recordings no earlier than 1 h after slice preparation.

### 4.4. Electrophysiology

Slices were fully submerged and continuously perfused (1 mL/min) with carbogen-saturated ACSF at RT. Slices were viewed with an Olympus BX51WI upright microscope (Olympus, Shinjuku, Japan) fitted with a 4x air (Plan N, NA 0.1; Olympus, Shinjuku, Japan) and 40x water-immersion (LUMPlan FI/IR, NA 0.8 w; Olympus, Shinjuku, Japan) objective and imaged with a CCD camera (XM10R, Olympus, Shinjuku, Japan).

Electrical signals were acquired at 10 kHz for mEPSC recordings and 50 kHz for all other recordings using an EPC10 amplifier (HEKA, Reutlingen, Germany), connected to a probe and PC. Electrical signals were recorded with the help of Patchmaster software (HEKA, Germany). Liquid junction potential was not adjusted. Ten micrometers of SR95531 hydrobromide (Biotrend, Germany) were added to the ACSF for all recordings. For mEPSC recordings, 1 µM TTX (Biotrend, Cologne, Germany) and 50 µM APV (Biotrend, Cologne, Germany) were added to the ACSF. For the analysis of NMDAR decays, 50 µM CNQX (Biotrend, Cologne, Germany) and 10 µM SR95531 hydrobromide (Biotrend, Cologne, Germany) were added to the ACSF.

Extracellular stimulation of pyramidal cells in layer 5 of the somatosensory cortex was performed by placing a stimulation electrode (chlorinated silver wire inside a borosilicate glass capillary filled with ACSF) in layer VI in close proximity to the patched neuron. The stimulus was generated by a stimulus isolator (WPI, Sarasota, FL, USA) connected with the EPC10 amplifier, which was connected to the stimulation pipette and the output was triggered by the Patchmaster software (version v2x90).

Extrasynaptic NMDARs were analyzed as described previously [107]. Briefly, nucleated patches were pulled from pyramidal cells in the somatosensory of acute brain slices. Patches were placed in front of a theta glass mounted onto a piezo translator (PI, Karlsruhe, Germany) that allowed the application of 1 mM glutamate for 1 ms. Application pipettes were tested by perfusing solutions with different salt concentrations through the two barrels onto open patch pipettes and recording current changes with 1 and 100 ms transitions of the application pipette. Only application pipettes with 20–80% rise times below 120 μs and with a reasonable symmetrical on- and offset were used. The application solution contained (in mM): 135 NaCl, 10 HEPES, 5.4 KCl, 1.8 CaCl_2_, 5 glucose, 0.01 CNQX, and 0.01 glycine (pH 7.2) without and without 1 mM glutamate.

### 4.5. Morphological Analysis

Pyramidal cells of the somatosensory cortex were filled with an intracellular solution (see above) containing 0.1–0.5% biocytin (Sigma Aldrich, St. Louis, MO, USA) through the patch-pipette during electrophysiological recordings. Subsequently, slices were fixed in 4% Histofix solution (Carl Roth, Karlsruhe, Germany) for several days, washed in 1xPBS (phosphate buffered saline) and permeabilized in 0.2% PBS-T (1xPBS with 0.2% Triton). A Streptavidin-coupled fluorescent dye (Alexa Fluor^TM^ 594 streptavidin, life technologies, Carlsbad, CA, USA) was incubated overnight at a dilution of 1:1000, in order to bind to biocytin. Acute slices obtained from 5xFAD and WT mice were additionally stained with an Alexa488-coupled 6E10 Antibody (Covance, New Jersey, DE, USA) at a dilution of 1:400 for Aβ staining. After washing in 1xPBS, slices were mounted in ProLong Gold Antifade (life technologies, USA). Z-stacks of selected dendrites of pyramidal cells were imaged with a fixed-stage Leica TCS SP5 II microscope (Leica, Germany) and the Leica LAS AF Lite Software (Leica, Wetzlar, Germany) with a 63x oil-immersion objective (Leica, Germany) with the following parameters: voxel size x/y = 0.08 µm, z = 0.168 µm. Dendritic spines were counted semi-automatically with the help of Neuronstudio (CNIC, Mount Sinai School of Medicine, USA). Amira (Thermo Fisher Scientific, Waltham, MA, USA) was used for blind deconvolution to improve image quality for spine analysis.

### 4.6. Analysis and Statistics

mEPSCs were analyzed with a template matching algorithm using the mini-analysis plugin of the Clampfit software (Molecular Devices, San José, CA, USA). Intrinsic electrophysiological properties, firing pattern analysis, as well as decay and deactivation analyses were performed with IGOR Pro (WaveMetrix, Lake Oswego, OR, USA) containing the Patcher’s Power Tools and Neuromatic analysis package (MPI for biophysical chemistry, Germany and Jason Rothman [108]). The weighted decay and deactivation time constants were calculated as τw = (τf x af) + (τs x as), where af and as are the relative amplitudes of the fast (τf) and slow (τs) exponential components.

Microsoft Office Excel (Microsoft, USA) and GraphPad Prism (GraphPad 6 software, San Diego, CA, USA) were used for data processing and statistical calculations. Datasets were tested for statistical significance with t-test (for normally distributed data), Mann–Whitney (MW; for not-normally distributed data) or Kruskal–Wallis (followed by Dunn’s posttest) tests. Data are shown as median ± interquartile ranges (IQR). *p* values < 0.05 were considered statistically significant (* = *p* < 0.05, ** = *p* < 0.01, *** = *p* < 0.001). All figures were prepared with Adobe Illustrator CS5.5 (Adobe, Mountain View, CA, USA).

## Figures and Tables

**Figure 1 ijms-22-06298-f001:**
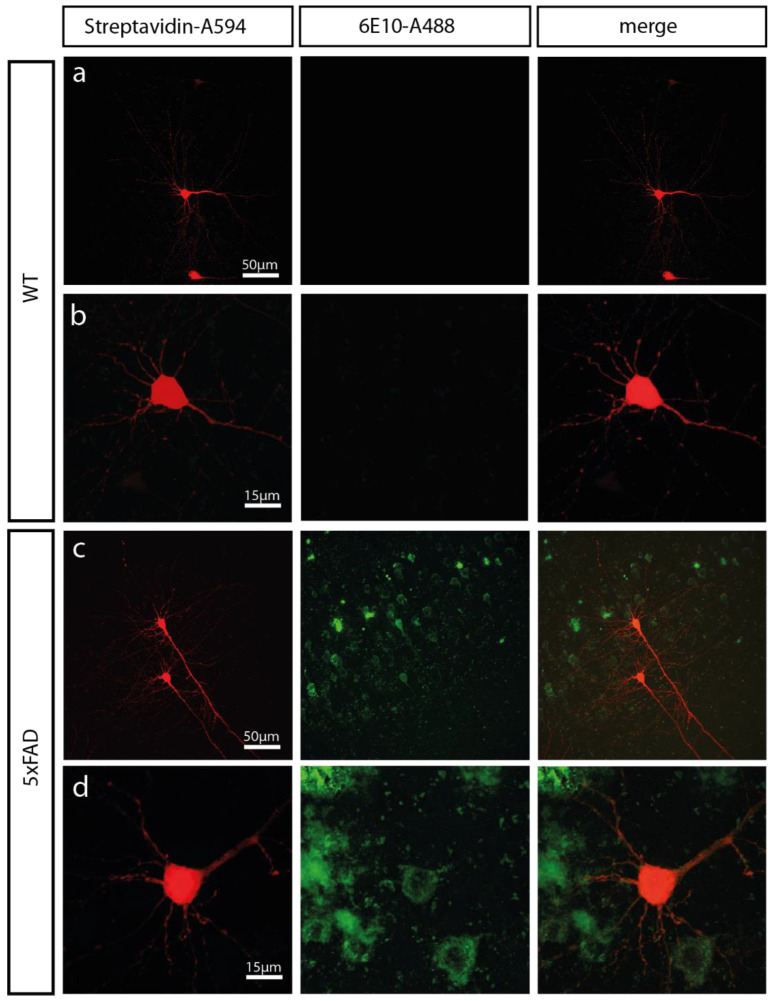
The somatosensory cortex of 6-months-old 5xFAD mice displays abundant amyloid beta (Aβ) load. Low (**a** + **c**) and high-magnification (**b** + **d**) confocal images of biocytin-filled pyramidal cells in the somatosensory cortex of a 6-months-old 5xFAD mouse and wildtype (WT) mouse. Biocytin-filled neurons were detected by fluorescently labelled streptavidin (Streptavidin-A594; left panel). Aβ protein was detected with the 6E10 antibody coupled to A488 (6E10-A488; middle panel). Merges of both channels are shown in the right panel.

**Figure 2 ijms-22-06298-f002:**
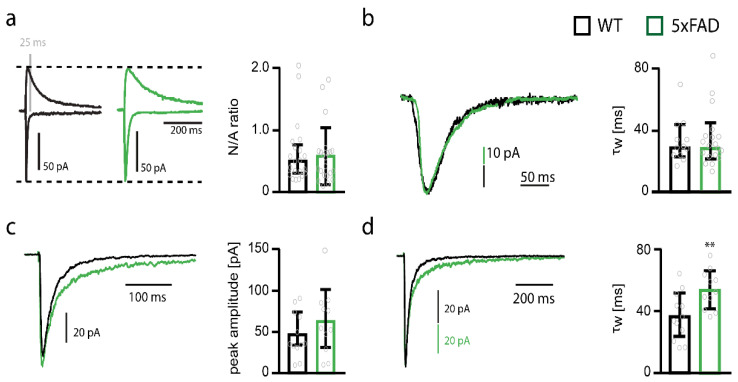
Deactivation time constant of extrasynaptic NMDAR currents increases in somatosensory cortex pyramidal cells of 5xFAD mice. (**a**) Example traces of NMDAR and AMPAR-mediated currents and bar graph of the NMDA/AMPA ratios (N/A ratios) of pyramidal cells from wildtype (WT) (black, *n* = 21) and 5xFAD (green, *n* = 22) mice. (**b**) Example traces and bar graph of the decay weighted tau (τ_w_) of NMDAR-mediated currents of pyramidal cells from WT (*n* = 12) and 5xFAD (*n* = 20) mice. (**c**) Example traces of extrasynaptic NMDAR-mediated currents recorded by ultra-fast application of glutamate onto nucleated patches obtained from somatosensory cortex pyramidal cells of WT (black) and 5xFAD (green) mice. Peak amplitudes (WT *n* = 12 vs. 5xFAD *n* = 11) of NMDAR-mediated currents are shown in the bar graph. (**d**) Example traces of normalized extrasynaptic NMDAR-mediated currents recorded by ultra-fast application of glutamate onto nucleated patches obtained from somatosensory cortex pyramidal cells of WT (black) and 5xFAD (green) mice. Weighted time constants (τ_w_) of NMDAR-mediated current deactivation are shown in the bar graph (WT *n* = 12 vs. 5xFAD *n* = 11). Data in bar graphs are shown as median ± interquartile range. Empty grey circles depict single data points. *t*-test: ** *p* < 0.001.

**Figure 3 ijms-22-06298-f003:**
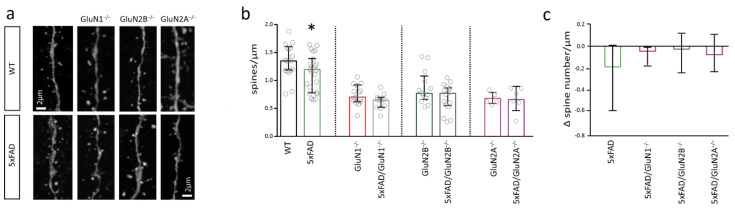
Loss of dendritic spines in 5xFAD mice depends on NMDARs. (**a**) Sample images of dendrites containing spines of pyramidal cells of the somatosensory cortex of wildtype (WT) and 5xFAD mice with and without deletion of GluN1, GluN2B, and GluN2A. (**b**) Bar graph showing the number of spines per µm in neurons of WT (*n* = 17) and 5xFAD (*n* = 23) mice (left), in neurons with GluN1 deletion (middle left) (*n* = 15 GluN1^−/−^ vs. *n* = 10 5xFAD/GluN1^−/−^), GluN2B deletion (middle right) (*n* = 13 GluN2B^−/−^ vs. *n* = 15 5xFAD/GluN2B^−/−^) or GluN2A deletion (right) (*n* = 5 GluN2A^−/−^ vs. *n* = 6 5xFAD/GluN2A^−/−^). (**c**) Bar graph showing the Δ spine number/µm in pyramidal cells from 5xFAD mice when compared to the respective control mouse (WT, GluN1^−/−^, GluN2B^−/−^, GluN2A^−/−^), quantified from data in (**b**). Data in bar graphs are shown as median ± interquartile range. Empty grey circles depict a single data point. *t*-test: * *p* < 0.05.

**Figure 4 ijms-22-06298-f004:**
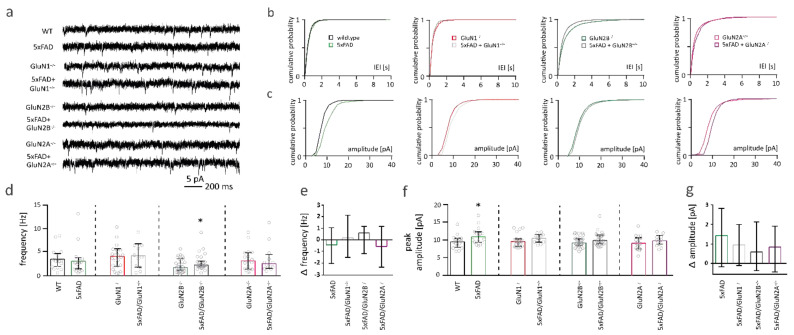
Functional synapses are only mildly affected in pyramidal cells of the somatosensory cortex from 5xFAD mice. (**a**) Sample traces of miniature excitatory postsynaptic current (mEPSC) recordings. (**b**) Cumulative distribution plots of the median inter-event-intervals (IEI) of mEPSCs of pyramidal cells of the somatosensory cortex. (**c**) Cumulative distribution plots of the median amplitudes of mEPSCs. (**d**) Bar graphs of mEPSC frequencies neurons of wildtype (WT) (*n* = 16) and 5xFAD (*n* = 16) mice (left), with deletion of GluN1 (middle left) (*n* = 22 GluN1^−/−^ vs. *n* = 11 5xFAD/GluN1^−/−^) or GluN2B (middle right) (*n* = 32 GluN2B^−/−^ vs. *n* = 30 5xFAD/GluN2B^−/−^) or GluN2A (right) (*n* = 17 GluN2A^−/−^ vs. *n* = 12 5xFAD/GluN2A^−/−^). (**e**) Bar graph showing the change in mEPSC frequency in pyramidal cells from 5xFAD mice when compared to the respective control mouse (WT, GluN1^−/−^, GluN2B^−/−^, GluN2A^−/−^), quantified from data in (**d**). (**f**) Bar graphs of peak amplitudes of mEPSCs from WT and 5xFAD mice (left) with GluN1 (middle left) or GluN2B (middle right) or GluN2A (right) deletion. (**g**) Bar graph showing the change in mEPSC amplitude in pyramidal cells from 5xFAD mice when compared to the respective control mouse (WT, GluN1^−/−^, GluN2B^−/−^, GluN2A^−/−^), quantified from data in (**f**). Data in bar graphs are shown as median ± interquartile range. Empty grey circles depict single data points. Mann–Whitney test/Kruskal–Wallis test: * *p* < 0.05.

**Figure 5 ijms-22-06298-f005:**
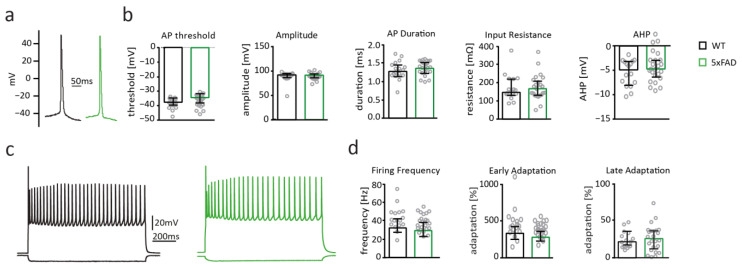
5xFAD somatosensory cortex pyramidal cells display unaltered intrinsic electrophysiological properties and firing behavior. (**a**) Example traces of action potentials recorded from pyramidal cells of the somatosensory cortex of wildtype (WT) (black) and 5xFAD (blue) mice. (**b**) Bar graphs displaying action potential (AP) threshold, AP amplitude and duration, input resistance, and afterhyperpolarization (AHP) of pyramidal cells from WT (black; *n* = 17) and 5xFAD mice (green; *n* = 24). (**c**) Example traces of action potential firing patterns recorded from somatosensory cortex pyramidal cells of WT (black) and 5xFAD (green) mice. (**d**) Bar graphs of firing frequency, early and late adaption of pyramidal cells from WT (black), and 5xFAD (green) mice. Data in bar graphs are shown as median ± interquartile range. Empty grey circles depict single data points.

**Figure 6 ijms-22-06298-f006:**
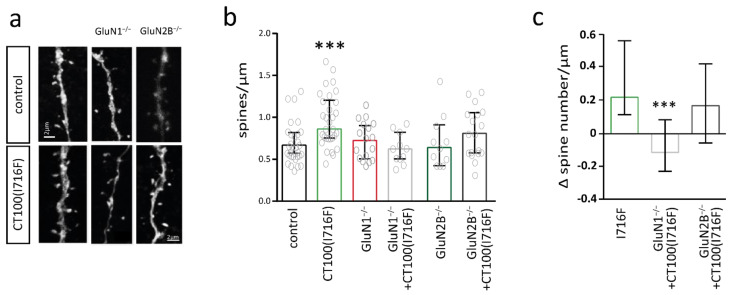
Increase in dendritic spine number in pyramidal cells of the somatosensory cortex of amyloid beta (Aβ)-overexpressing mice is mediated by the GluN1 subunit. (**a**) Sample images of dendritic spines of pyramidal cells of the somatosensory cortex without (control) and with CT100(I716F)-overexpression without (left) and with deletion of GluN1 (middle) and GluN2B (right). (**b**) Bar graph showing the number of spines per µm of control (*n* = 31) and CT100(I716F)-overexpressing (*n* = 31) neurons (left) with GluN1 deletion (middle) (*n* = 17 GluN1^−/−^ vs. *n* = 11 CT100(I716F)/GluN1^−/−^) or GluN2B deletion (right) (*n* = 11 GluN2B^−/−^ vs. *n* = 8 CT100(I716F)/GluN2B^−/−^). (**c**) Bar graph showing the change in spine number in pyramidal cells from CT100(I716F)-overexpressing neurons when compared to the respective control neurons (control, GluN1^−/−^, GluN2B^−/−^) quantified from data in (**b**). Data in bar graphs are shown as median ± interquartile range. Empty grey circles depict single data points. *t*-test; Kruskal–Wallis-test: *** *p* < 0.001.

**Figure 7 ijms-22-06298-f007:**
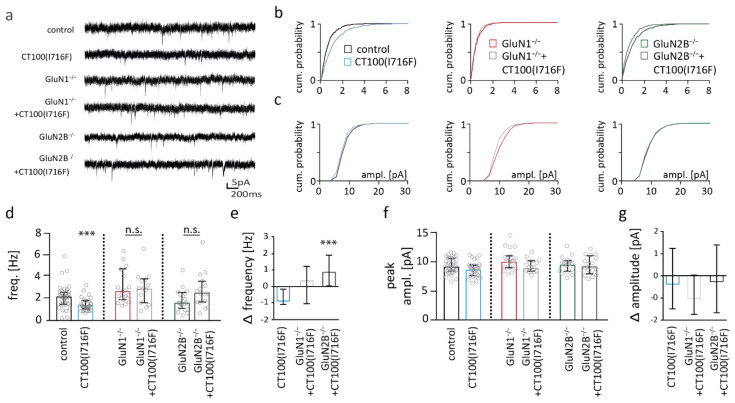
Reduction in miniature excitatory postsynaptic current (mEPSC) frequency is NMDAR-dependent in somatosensory cortex pyramidal cells of amyloid beta (Aβ)-overexpressing neurons. (**a**) Sample traces of mEPSC recordings. (**b**) Cumulative distribution plots of the median inter-event-intervals (IEI) of mEPSCs from pyramidal cells of the somatosensory cortex. (**c**) Cumulative distribution plots of the median amplitudes (IEI) of mEPSCs. (**d**) Bar graphs of mEPSC frequencies of control (*n* = 35) and CT100(I716F)-overexpressing (*n* = 31) neurons without (left) and with deletion of GluN1 (middle) (*n* = 19 GluN1^−/−^ vs. *n* = 14 CT100(I716F)/GluN1^−/−^) or GluN2B (right) (*n* = 18 GluN2B^−/−^ vs. *n* = 16 CT100(I716F)/GluN2B^−/−^). (**e**) Bar graph showing the change in mEPSC frequency in pyramidal cells from CT100(I716F)-overexpressing neurons when compared to the respective control neurons (control, GluN1^−/−^, GluN2B^−/−^), quantified from data in (**d**). (**f**) Bar graphs of peak amplitude of mEPSCs from control and CT100(I716F)-overexpressing neurons without (left) and with deletion of GluN1 (middle) or GluN2B (right). (**g**) Bar graph showing the change in mEPSC amplitude in pyramidal cells from CT100(I716F)-overexpressing neurons when compared to the respective control neurons (control, GluN1^−/−^, GluN2B^−/−^), quantified from data in (**f**). Data in bar graphs are shown as median ± interquartile range. Empty grey circles depict single data points. Mann–Whitney test/Kruskal–Wallis test: *** *p* < 0.001.

## Data Availability

The data presented in this study are available on request from the corresponding author.

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
