# Peer review of "Amyloid Beta-Mediated Changes in Synaptic Function and Spine Number of Neocortical Neurons Depend on NMDA Receptors"

_ijms, 2021, doi:10.3390/ijms22126298_

Round 1

Reviewer 1 Report

The manuscript entitled Regulation and role of NMDA receptors in cortical neurons of 2 Alzheimers’ disease model mice with amyloid-beta over-ex-3 pression describe theNMDA role in the AD pathophysiology, putting the attention in the mechanism underlying Aβ-mediated toxicity. The most important results observed by Michaela K. Back et al. have been the demonstration that the virus-mediated A_β-overexpression induce an upregulation of the extrasynaptic GluN2B-containing NMDARs, altering the decay and deactivation kinetics. And that Aβ-mediated change in spine number depended on the presence of NMDARs. The manuscript is well written, the afforded data is promising and interesting and well discussed. However, there are some points to be revised.

-In Figure 1 the authors say that the Aβ-staining was found mainly intracellularly in neurons of the somatosensory cortex of six-months-old 5xFAD mice accompanied by a smaller amount of extracellular Aβ plaques. However, the streptavidin staining show that not only biocytin-filled neurons of 5xFAD mice but also WT mice contain Aβ-staining. It would be nice if the authors could quantify the green fluorescence giving a plot data to consolidate the observed result.

-In Figure 3 the authors show the loss of spines in pyramidal cells of the somatosensory cortex of WT and 5xFAD mice by delecting GluN1 and GluN2B genes. Due to the continuous comparation between GluN2A and  GluN2B, it would be interesting for the lector to receive the data obtained by delecting GluN2A subunit, comparing different NMDA receptors.

-It would also be nice that the authors could demonstrate that the virus injection per se do not induce any important effect to study. An important negative control could be the injection with an empty vector.

-The authors demonstrate that it is necessary a deletion of GluN1 or GluN2B almost three month to detect affection in spiny number. It would be nice that the authors could discuss the differences induced in the brain three months after NMDA delection compared to three weeks.

-There are some sentences that should be better explained because is difficult to understand the message the authors desire to show. An example is this sentence in the abstract: NMDARs were required for both A_β-mediated changes in spine number.

Author Response

General remark: We thank the reviewer for his/her positive judgement of our work and are glad that he/she finds it “promising”, “interesting” and “well discussed”.

Point 1: In Figure 1 the authors say that the Aβ-staining was found mainly intracellularly in neurons of the somatosensory cortex of six-months-old 5xFAD mice accompanied by a smaller amount of extracellular Aβ plaques. However, the streptavidin staining show that not only biocytin-filled neurons of 5xFAD mice but also WT mice contain Aβ-staining. It would be nice if the authors could quantify the green fluorescence giving a plot data to consolidate the observed result.

Reply to point 1: Indeed, a weak signal can be seen in the green channel. However, this signal was due to bleed through from the red channel (i.e. from the biocytin-staining). Consistently, no other neuron than the biocytin-filled neuron showed green signal intracellularly. Similarly, the extracellular green dots that colocalized with red dots were observed only in the vicinity of biocytin-filled neurons and resulted from biocytin blown onto the acute brain slice during patching procedure. In contrast, the specific 6E10-A488 signal that is observed on 5xFAD brain slices is not colocalizing with red signal and is ubiquitous and not only in the vicinity of biocytin-filled neurons.

To reduce this ambiguity, we now imaged brain slices of WT and 5xFAD mice using improved imaging settings at the confocal microscope which separate green and red channel better than the previous settings. We also include a high magnification image of the biocytin-filled neuron. In addition, we filled neurons very carefully with biocytin in order to reduce spillover of biocytin onto the slice.

Point 2: In Figure 3 the authors show the loss of spines in pyramidal cells of the somatosensory cortex of WT and 5xFAD mice by delecting GluN1 and GluN2B genes. Due to the continuous comparation between GluN2A and GluN2B, it would be interesting for the lector to receive the data obtained by delecting GluN2A subunit, comparing different NMDA receptors.

Reply to point 2: We now included data of spine and mEPSC quantifications of neurons with GluN2A deletion in 5xFAD mice and changed Figures 3 and 4 accordingly. Loss of dendritic spines in pyramidal cells of the somatosensory cortex can also be prevented by deletion of the GluN2A subunit in six-month-old 5xFAD mice. Thus, all three subunits (GluN1, GluN2B and GluN2A) are involved in Aβ-mediated spine loss in the somatosensory cortex.

Point 3: It would also be nice that the authors could demonstrate that the virus injection per se do not induce any important effect to study. An important negative control could be the injection with an empty vector.

Reply to point 3: The explanation of controls in our manuscript was poorly described. We apologize for this misleading fact. Indeed, the control mice were injected with an “empty” rAAV, only coding for a fluorescence marker (tdTomato), as stated in the M&M section: “The following rAAVs were stereotactically injected trough a thin glass capillary into the somatosensory cortex (anteroposterior, -2.5 mm; mediolateral, +3.5 mm; dorsoventral, -0.5 mm according to Bregma): rAAV-CaMKII-tdTom (control cells),…”. However, we did not mention this fact anywhere else in the text. We now included a more detailed explanation of the controls in the M&M section: “Littermates were used as controls in all experiments. In experiments with virus-mediated Aβ overexpression, we used GluN1fl/fl and GluN2Bfl/fl mice injected with a tomato-expressing rAAV as controls (for detail see below). We verified before that the injection of the tomato and Cre-expressing rAAVs as well as the insertion of the loxP sites per se do not alter the expression of AMPA and NMDARs (own unpublished data). For experiments with the 5xFAD line, we injected the tomato-expressing rAAV in 5xFADxGluN1fl/fl and 5xFAD/GluN2Bfl/fl mice (termed 5xFAD in this paper) and littermate GluN1fl/fl and GluN2Bfl/fl mice (termed WT in this paper).“

Point 4: The authors demonstrate that it is necessary a deletion of GluN1 or GluN2B almost three month to detect affection in spiny number. It would be nice that the authors could discuss the differences induced in the brain three months after NMDA delection compared to three weeks.

Reply to point 4: This is indeed a very interesting point. We added additional information and suggestions on this issue in the discussion: “The interpretation of the requirement of NMDARs for the effect of Aβ on spine number is somewhat compromised by the fact that deletion of NMDARs alone strongly reduces spine number after NMDAR subunit deletion for 12 weeks in six-month old mice, but not after three weeks in three-month old mice. Thus, the duration of NMDAR absence and/or the age of the animal seem to be critical. Spines can be classified as transient (lifetime of minutes to hours) and persistent (lifetime of days to months) (Holtmaat et al., 2005; Berry, Nedivi, 2018). The largest fraction of spines are of the persistent type in the somatosensory cortex of adult mice (Grutzendler et al.; 2002; Holtmaat et al., 2005). The slow turnover explains therefore that spine loss is present 12 weeks but not three weeks after NMDAR deletion.

Point 5: There are some sentences that should be better explained because is difficult to understand the message the authors desire to show. An example is this sentence in the abstract: NMDARs were required for both A_β-mediated changes in spine number.

Reply to point 5: We apologize for mistakes in phrasing. We carefully re-read the manuscript and edited (in our finding) unclear sentences. The sentence “NMDARs were required for both A_β-mediated changes in spine number” was corrected toNMDARs were required for both Aβ-mediated changes in spine number and functional synapses”.

Reviewer 2 Report

Ref: ijms-1228882

Title: Regulation and role of NMDA receptors in cortical neurons of Alzheimers’ disease model mice with amyloid-beta over-expression (Journal: IJMS, MDPI)

Recommendation: Major review

Comments:

  1. Please, provide the detailed description of materials and methods (e.g. sex of the animals, detailed description of the mice models (wild and other control)).
  2. Please explain the therapeutic potential of the conducted research.
  3. I assume that only males were used in the research. Research should be conducted using both sexes.
  4. Please provide the dilutions of used antibodies.
  5. I would suggest to change the title. The current is too general and looks like the title of the review paper.

Author Response

Reviewer #2:  

Point 2.1: Please, provide the detailed description of materials and methods (e.g. sex of the animals, detailed description of the mice models (wild and other control)).

Reply to point 2.1: We apologize for incomplete informations in the materials and methods section. We now added the following information in the M&M section:

“Only female 5xFAD mice were used for experiments, since male and female 5xFAD show significant differences in Aβ plaque load (Bhattacharya et al., 2014). For experiments, in which Aβ overproduction was mediated via rAAVs, data obtained from female and male mice was pooled.

Littermates were used as controls in all experiments. In experiments with virus-mediated Aβ overexpression, we used GluN1fl/fl and GluN2Bfl/fl mice injected with a tomato-expressing rAAV as controls (for detail see below). We verified before that the injection of the tomato and Cre-expressing rAAVs as well as the insertion of the loxP sites per se do not alter the expression of AMPA and NMDARs (own unpublished data).

For experiments with the 5xFAD line, we injected the tomato-expressing rAAV in 5xFADxGluN1fl/fl and 5xFAD/GluN2Bfl/fl mice (termed 5xFAD in this paper) and littermate GluN1fl/fl and GluN2Bfl/fl mice (termed WT in this paper).”

Point 2.2: Please explain the therapeutic potential of the conducted research.

Reply to point 2.2: We added more information on the therapeutic potential of our research in the discussion part:

“Knowledge of the brain-region specific composition and contribution of the NMDAR subunits to AD pathophysiology could be important for the development of novel AD treatments. For example, antagonists that are specific for AD-relevant NMDAR compositions might be more efficient than memantine without the cost of stronger side effects. Antagonists that are more specific for triheteromeric NMDARs could be more beneficial than memantine for the treatment of AD patients considering the upregulation of triheteromeric NMDARs at extrasynaptic sites in 5xFAD somatosensory pyramidal cells.

We also added more information at the end of the discussion section:

Thus, early treatment with specific antagonists that acts on triheteromeric (GluN1/2A/2B) NMDARs in the neocortex might be more efficient and might have fewer side-effects than broad blockade of all NMDARs by memantine.”

Point 2.3: I assume that only males were used in the research. Research should be conducted using both sexes.

We used mice of both sexes for the experiments, in which Aβ production is induced by the infection of cells with an rAAV. For experiments in the 5xFAD mouse line, we used only female mice, since there are large sex-specific differences in the progression of pathophysiology and severity of amyloid beta plaque occurrence in this mouse model (Bhattacharya S, Haertel C, Maelicke A, Montag D. Galantamine slows down plaque formation and behavioral decline in the 5XFAD mouse model of Alzheimer's disease. PLoS One. 2014;9(2):e89454).

Mixing data from female and male mice would significantly increase data variability.  Thus, only very large changes in spine number or mEPSC frequency and amplitude would become significant and/or unreasonable high mouse numbers would be required to show statistically significant smaller differences. Thus, we believe there is good reason to focus only on one sex.

We now added the following information to the M&M section, in order to clarify this point:

Only female 5xFAD mice were used for experiments, since male and female 5xFAD show significant differences in Aβ plaque load (Bhattacharya et al., 2014). For experiments, in which Aβ overproduction was mediated via rAAVs, data obtained from female and male mice was pooled.”

Point 2.4: Please provide the dilutions of used antibodies.

Antibody dilutions were added to the corresponding sentences in the M&M section:

A Streptavidin-coupled fluorescent dye (Alexa FluorTM 594 streptavidin, life technologies, USA) was incubated overnight at a dilution of 1:1000, in order to bind to biocytin. Acute slices obtained from 5xFAD and WT mice were additionally stained with an Alexa488-coupled 6E10 Antibody (Covance, USA) at a dilution of 1:400 for Aβ staining.”

Point 2.5: I would suggest to change the title. The current is too general and looks like the title of the review paper.

The title of the manuscript was now changed to “Amyloid beta-mediated changes in synaptic function and spine number of neocortical neurons depend on NMDA receptors”.

Round 2

Reviewer 1 Report

I accept the revised version

Reviewer 2 Report

Due to the fact that the Authors have answered all my questions/doubts, I recommend the paper for publication.